# QSAR Models for Human Carcinogenicity: An Assessment Based on Oral and Inhalation Slope Factors

**DOI:** 10.3390/molecules26010127

**Published:** 2020-12-29

**Authors:** Cosimo Toma, Alberto Manganaro, Giuseppa Raitano, Marco Marzo, Domenico Gadaleta, Diego Baderna, Alessandra Roncaglioni, Nynke Kramer, Emilio Benfenati

**Affiliations:** 1Laboratory of Chemistry and Environmental Toxicology, Department of Environmental Health Sciences, Istituto di Ricerche Farmacologiche Mario Negri IRCCS, Via Mario Negri 2, 20156 Milan, Italy; cosimo.toma@marionegri.it (C.T.); giuseppa.raitano@marionegri.it (G.R.); marco.marzo@marionegri.it (M.M.); domenico.gadaeta@marionegri.it (D.G.); alessandra.roncaglioni@marionegri.it (A.R.); 2Institute for Risk Assessment Sciences (IRAS), Utrecht University, P.O. Box 80177, 3508 TD Utrecht, The Netherlands; N.I.Kramer@uu.nl; 3Kode Chemoinformatics s.r.l., 56125 Pisa, Italy; a.manganaro@kode-solutions.net

**Keywords:** cancer slope factor, in silico method, QSAR, prioritization

## Abstract

Carcinogenicity is a crucial endpoint for the safety assessment of chemicals and products. During the last few decades, the development of quantitative structure–activity relationship ((Q)SAR) models has gained importance for regulatory use, in combination with in vitro testing or expert-based reasoning. Several classification models can now predict both human and rat carcinogenicity, but there are few models to quantitatively assess carcinogenicity in humans. To our knowledge, slope factor (SF), a parameter describing carcinogenicity potential used especially for human risk assessment of contaminated sites, has never been modeled for both inhalation and oral exposures. In this study, we developed classification and regression models for inhalation and oral SFs using data from the Risk Assessment Information System (RAIS) and different machine learning approaches. The models performed well in classification, with accuracies for the external set of 0.76 and 0.74 for oral and inhalation exposure, respectively, and r^2^ values of 0.57 and 0.65 in the regression models for oral and inhalation SFs in external validation. These models might therefore support regulators in (de)prioritizing substances for regulatory action and in weighing evidence in the context of chemical safety assessments. Moreover, these models are implemented on the VEGA platform and are now freely downloadable online.

## 1. Introduction

Every day, people are exposed to numerous environmental chemical stressors that can have adverse health effects during their life. Exposure to toxic chemicals or mixtures comes from the environment, living places and workplaces, but diet, drugs and lifestyle are important concurrent sources as well [1,2,3,4]. Adverse effects include chronic diseases and cancer. Nowadays, cancer is a major public health issue with more than 3 million new cases per year in the European Union [5,6].

Experimentally, the carcinogenic potential of a substance is evaluated by long-term in vivo carcinogenicity studies with laboratory animals. The conventional test for carcinogenicity is the two-year rodent carcinogenicity bioassay as described by the Organization for Economic Co-operation and Development (OECD) Test Guidelines 451 and 453 [7,8,9]. Although the procedure is expensive and time-consuming, animal models are still the most widely used method of investigation. In the last decade, the validity of the rodent bioassay was debated because of uncertainty related to extrapolating results to humans and ethical concerns about the numbers of animals needed [6]. Various non-animal methods have recently been proposed as alternative or complementary methods to assess carcinogenicity with the aim of reducing animal testing, time and cost of the evaluation. These methods include in vitro bioassays (such as cell transformation assays and toxicogenomics) and in silico methods, such as (Q)SARs models and expert systems [10,11,12,13,14,15,16,17,18,19,20].

Most of the *in silico* models for carcinogenicity are classifier tools used to predict whether or not chemicals are carcinogens in animal models [15,21,22,23,24]. Only a few continuous models are used to quantitatively assess carcinogenicity, specifically, to predict the potency dose in vivo as the endpoint (TD50) [25,26,27,28,29,30,31,32]. Most of these models are already implemented in license-based or freely available software tools (Appendix A).

To our knowledge, no model has yet been developed for oral and inhalation slope factors (SFs) used in human quantitative risk assessment (HRA) of environmental pollutants. The SF is the upper-bound estimate of the slope of the dose–response curve in the low-dose region for carcinogens and is used to assess the increase over a lifetime in incidence of cancers in humans from oral or inhalation exposure to a dose of a carcinogenic chemical [33,34,35,36]. In the HRA framework, the cancer risk for each chemical (CR or Incremental Lifetime Cancer Risk, ILCR) is calculated using the chronic daily intake (CDI, mg/kg-day) and the slope factor (SF, (mg/kg-day) ^−1^); the SF provides the chemical-specific carcinogenic potency. With these two values, the cancer risk (CR, dimensionless) is obtained by multiplying them, as in Equation (1):CR = CDI × SF(1)

Here we propose an integrated in silico approach for the qualitative and quantitative assessment of chemical carcinogenic potency, which includes classification and quantitative models for inhalation and oral human carcinogenicity based on slope factors.

## 2. Results

We developed both classification and regression models for carcinogenicity expressed as oral or inhalation slope factors. Data were collected from the Risk Assessment Information System (RAIS) Toxicity values database (https://rais.ornl.gov). For each exposure route (oral or inhalation), chemicals with a defined value for SF were considered carcinogenic, while compounds with no value were considered non-carcinogenic. This binary dataset was used to develop the classification models; however, the dataset including the SF values to describe carcinogenic potency was used for the regression models. Thus, in our proposed approach, the classifier models indicate if the substance is carcinogenic or not, and the regression model should be used in cascade to assess the substance’s potency if it is labelled as carcinogenic.

### 2.1. Classification Models

Binary classification models were built by the Classification and Regression Tree (CART) modelling approach. CART models for inhalation and oral carcinogenicity performed well for sensitivity and specificity. The structure of inhalation and oral CART models are included in the Appendix A. In order to increase access to the models, we implemented them within the platform VEGA (Virtual models for property Evaluation of chemicals within a Global Architecture, www.vegahub.eu), our online, freely available platform that contains a series of QSAR models for regulatory purposes. There are negligible differences between the original models in CART and those in VEGA. Balancing the dataset between models led, as expected, to lower sensitivity for the inhalation model than for the oral model (Table 1).

### 2.2. Regression Models

We developed the regression models using descriptor-based multi-layer perceptron–artificial neural networks (MLP–ANNs). In the proposed strategy to assess carcinogenicity, the regression model should be run when the classification model indicates carcinogenicity. For each regression model, the performance is reported as a determination coefficient (r^2^), root-mean-square error (RMSE), mean absolute error (MAE) and percentage of predicted compounds (this percentage is called coverage of the model). We split the substances into two approaches. As detailed in the Material and Methods section, in one case (split A) we used a test set (TeS) to select the best model, and then an external validation set (ES) was used to evaluate the performance of new substances. In approach B, there was only a training and validation set, and the model selection was done using the 10-fold cross-validation method, using substances from the training set. Performances are reported in Table 2 and Table 3 for the two splitting approaches, with values for the training set (TrS), TeS and ES.

Split scheme A returned similar results for both OSF and ISF. The predictive power of both models was confirmed on both validation sets, with r^2^ values from 0.70 to 0.65 on the TeS, and from 0.57 to 0.51 on the ES. Coverage of the models on the two validation sets was always greater than 80%. Models derived from split scheme B showed similar results, though performance was slightly better for the ISF model that returned an r^2^ of 0.65 on the ES, while the OSF model returned an r^2^ of 0.52. Moreover, the coverage on the TeS for the OSF was lower than 80%. Thus, split scheme A can be considered preferable for model OSF, while split scheme B gave better results for model ISF.

One disadvantage of DRAGON descriptors is that they cannot be used to develop a completely free and open source QSAR model software, even though they are widely used and robust. There are some examples of QSAR models retrained with DRAGON-like descriptors that give similar results [37,38]. With a view of implementing all of the developed models in the VEGA platform [39], we retrained them using the best scheme for each endpoint (scheme A for OSF and scheme B for ISF).

The model for OSF was replicated without modifications because VEGA already has the same descriptors as DRAGON models; however, the selected descriptors were not available as DRAGON-like descriptors for the inhalation models, so retraining the model led to different performances.

Table 4 shows the statistics of the models implemented in VEGA. The implementation of OSF with DRAGON gave higher r^2^ in training (TrS = 0.709, TeS = 0.708) than VEGA did (TrS + Te = 0.62), which gave a better performance in ES (0.569 vs. 0.839). However, r^2^ values for the TrS and ES of ISF (TrS = 0. 745, ES = 0.647) were both certainly higher than with VEGA implementation (TrS = 0.586, ES = 0.566). Within the feature selection, we reduced the number of descriptors, using tools such as genetic algorithms, as described in the Materials and Methods section. One possible explanation is that a genetic algorithm using a larger number of descriptors in DRAGON had more starting combinations and a greater chance of selecting the best pool of descriptors.

## 3. Discussion

Here we propose classification and regression models for the carcinogenicity risk assessment of organic chemicals. Classification models are used to detect potential carcinogens and assume that negatively predicted compounds are non-carcinogenic. Meanwhile, the regression models quantify the potency of each chemical as a slope factor. We developed models for both inhalation and oral carcinogenicity.

The carcinogenicity models performed well: accuracies in the test set were 0.76 and 0.74 for oral and inhalation models, respectively, and r^2^ values were 0.57 and 0.65 in the regression models, respectively, for oral and inhalation SFs in ES.

Our results suggest that these models could be useful to support regulators in chemical safety assessments, providing information not only on the carcinogenic potential of chemicals but also as a measure of their potency. This latter information is fundamental to establish threshold concentrations of each chemical carcinogen, and it also gives a quantitative estimate of the risk of adverse health effects in exposed recipients.

### 3.1. Focusing on Selected Descriptors

Even though the models were trained with different split schemes and different datasets, some descriptors were selected in more than one model, and others came from the same descriptor block (Appendix A).

The cyclomatic number (nCIC), for example, was selected for both classification and regression slope factors and refers to the number of rings. The descriptor is related to the high carcinogenicity potency and to the large number of rings in the same molecule and is typically seen in polycyclic aromatic hydrocarbons (PAHs), which are carcinogenic through the formation of epoxides [40].

Another important class of descriptors selected for the regression models relates to the presence of chlorine. Several non-genotoxic mechanisms are influenced by the presence of halogens. For example, polychlorinated biphenyl (PCB) interaction with the aryl hydrocarbon receptor (AHR) plays a major role in breast cancer. It has also been reported that the affinity is related to the planar conformation of the molecule [41,42,43]. Though the developed model does not take account of 3D information, the position of halogens on the ring (B02[Cl-l], B04[O-Cl], B07[Cl-Cl], B08[Cl-Cl], B08[Cl-Cl] and F04[O-Cl]) heavily influences the planarity of the molecule. This is well known for dioxin-like PCBs [44].

Descriptors like nRNNOx (number of nitroso groups) and nN-N identify several indirect alkylating agents, such as hydrazine or N-nitroso groups, that can form DNA adducts after metabolic activation.

### 3.2. Usefulness of the Model

The classification models can help spot uncertain data from the original dataset. For example, vinyl chloride, 1,3-butadiene and chloromethane are predicted as non-carcinogens by the oral classification model, even though they have an oral slope factor value. If we look at the origin of the data, we see that the results for these three substances were extrapolated from inhalation tests on rats because they have a gaseous state at 20 °C [45].

The same information holds with the inhalation classification model. Thiourea, methylthiouracil and acetamide are classified as non-carcinogenic by the inhalation classification model, but there is an inhalation slope factor for them. The inhalation risk arises only if the particles are smaller than 5 μm [46,47]. The three misclassified substances have a particle size reported in the OECD QSAR toolbox [48] above this threshold, making them unlikely to have a carcinogenic effect via inhalation.

### 3.3. Model Integration

The use of these models should follow a hierarchical pipeline. Since the regression models are based on a subset of compounds included in the classification dataset, a smaller chemical space will be covered. For this reason, substances should first be screened with the classification-based model in order to evaluate any carcinogenic effect, before then evaluating the potency with the relative regression model. This suggested workflow is outlined in Figure 1.

## 4. Materials and Methods

### 4.1. Datasets and Data Curation

Regression and classification models were developed using data from the RAIS Toxicity values database [49]. Data cover different pollutant categories including organic and inorganic compounds, such as dioxins, PAHs, pesticides and metals frequently found in contaminated sites. We retrieved 1110 and 990 values for the oral slope factor (OSF, (mg/kg-day)^−1^) and for the inhalation unit risk (IUR, µg/m^3^), respectively.

In accordance with the United States Environmental Protection Agency (US EPA) [50], IUR data were converted to inhalation slope factor (ISF) using the formula
ISF = (IUR ∗ BW ∗ CF)/IR(2)
where IUR = inhalation unit risk [(µg/m^3^) ^−1^], BW = average body weight [70 kg], IR = inhalation rate [20 m^3^/day] and CF = conversion factor [1000 µg/mg]. Both OSF and ISF values were then converted in logarithmic units for modelling purposes.

Canonical simplified molecular-input line-entry systems (canonical SMILES) were retrieved for each chemical from JChem for Office [51] and ChemID plus [52], and chemicals showing incongruency between the various sources were rejected. Most QSAR models cannot handle inorganic compounds, metals and metal complexes or organic salts, so data related to these compounds or mixtures were rejected. Ionized structures were neutralized and counterions eliminated. The datasets were further checked for duplicates.

Chemicals with a defined value (in our case SF) were considered carcinogenic, and compounds with no value were considered non-carcinogenic.

The final datasets for the classification models included 745 and 750 compounds, respectively, for OSF and ISF. For the regression models, only compounds with continuous data were used for modelling. This led to two final datasets comprising 315 compounds with OSF data and 263 with ISF data. Datasets are available on Zenodo [53] and also in the Appendix A.

### 4.2. Classification Models

We applied the same modelling workflow to OSF and ISF datasets. 2D molecular descriptors were calculated using DRAGON version 7.0.6 [54]. All available descriptors were selected, then pruned within DRAGON, removing those with missing values, constant and semi-constant values and redundant descriptors (those with a pairwise correlation over 0.95 with another descriptor).

For the training/test split, we used constitutional and ring descriptor blocks, together with the experimental class value, as input for principal component analysis (PCA). The first principal component (PC) was used to rank the compounds, then a venetian blind approach was used to split training and test set compounds in an 80–20% ratio.

An in-house tool developed in the R statistical platform [55] was used to select the best descriptors set and size to be employed for the final model. The approach was based on a forward selection technique, a well-known general strategy previously used by our group to build models for toxicological endpoints [56,57]. In this approach, the descriptor leading to the best model was added at each iteration, starting from the descriptor most closely correlated with the experimental data, until the final number of 25 descriptors. Models were built with Linear Discriminant Analysis (LDA) modelling and applied with a bootstrap cross-validation approach (100 iterations). The fitness function was calculated for each model as a linear combination of the means for accuracy, sensitivity and specificity from the models built in each bootstrap iteration. This function was used to select the best descriptor to be added in the process to proceed to the next iteration. The set of descriptors with the best cross-validation values was used for the final model.

The “best” values were defined on the basis of their trend: by progressively adding descriptors to the model, cross-validation performance increases up to a plateau, meaning that the optimal number of descriptors has been reached, and adding further descriptors would lead to over-fitting.

The optimal set of descriptors was used to build a CART model in the R statistical environment (using the “rpart” package [58]) for both datasets to improve the performance of the naïve LDA approach used in the first step. The CART modelling implemented in R includes an internal cross-validation to reduce the complexity of the model; as a result, the final trees contain fewer descriptors than the set provided as input (while still leading to better performance than with the LDA models).

### 4.3. Regression Models

Two different split schemes were applied on the OSF and ISF datasets (Figure 2). In split scheme A, 10% of the entire dataset was randomly extracted from the ES for external validation. The modeling set, consisting of the remaining chemicals, was split into a TrS and a TS [59] containing 80% and 20% of the modeling set, respectively. In split scheme B, only two datasets were created: TrS and ES, containing 80% and 20% of the entire dataset, respectively.

For both split schemes, uniform distribution of the endpoint values among the subsets was ensured by applying an activity sampling method: compounds were sorted on the basis of their activity and divided into bins of equal size in terms of activity range. For each bin, chemicals were randomly assigned to datasets based on the percentages [60,61,62]. Appendix A reports the size of the datasets with the two split schemes, OSF and ISF.

As for the classification models, molecular descriptors were calculated for each compound with DRAGON 7 software, and the same pruning procedure was applied for the regression models.

Features were selected with the “gaselect” R package [63] that implements a partial least-squares genetic algorithm (PLS-GA) and repeated double cross-validation [64] for statistical analysis of subsets of descriptors. The following settings were applied for the PLS-GA: initial population 2000; number of iterations 5000; minimum number of variables 5; maximum number of variables 12. Optimal subsets of descriptors returned by the final iteration of the run were used for model derivation, using r^2^ as a fitness function.

Regression models were derived from each optimal subset of descriptors with multi-layer perceptron–artificial neural networks (MLP–ANNs) [65], as implemented in KNIME software. The MLP node is an implementation of the RPROP algorithm for multilayer feed-forward networks. This is a multi-layer perceptron trained with backpropagation that performs a local adaptation of the weight-updates according to the behavior of the error function. This solution has been found useful to overcome the inherent disadvantages of pure gradient-descent. In this algorithm, weights near the input layer have an equal chance to grow and learn as weights near the output layer. In addition, the descriptors were normalized in a range from 0 to 1 to be used in the algorithm.

MLP–ANNs were trained over 100 iterations and had a standard architecture formed by one input layer (with the same number of nodes as input descriptors), one hidden layer (with 10 neurons) and one output layer.

The applicability domain (AD) [66,67,68] was defined in order to identify less reliable predictions that were more likely to be wrong. The AD was evaluated by PCA. TrS chemicals were projected on a chemical space defined by the first two principal components (PCs) calculated using model descriptors. TeS and ES chemicals whose PC1 or PC2 values fell outside the range defined by the 5th and 95th percentiles of the distribution for TrS compounds were considered outside the AD and excluded from the statistical analysis of the models. This restrictive approach was preferred to avoid the inclusion of underrepresented areas within the chemical space defined by the first two PCs. The AD was also defined using the standardization approach [69].

Models for OSF and ISF were derived for each of the two split schemes. The selection of the best models was different based on the splitting scheme applied. In scheme A, models were ranked according to their r^2^ value on the TeS, without considering compounds outside the AD. The single best-performing OSF and ISF models were then evaluated for their external predictivity on the ES. In scheme B, single best models were selected according to internal performance (i.e., r^2^ in 10-fold cross-validation) and then were evaluated for their external predictivity on the TeS.

All the models were then implemented in VEGA [39,70]. The OSF model has the same descriptors as the original DRAGON models since the VEGA engine is already able to calculate them. After recalculation of the descriptors with VEGA, the model was retrained using TrS + TeS, then validated on ES. Since selected descriptors for the inhalation regression model were not available as free descriptors in the VEGA platform, it was necessary to replicate the model using the same approach but with the descriptors already implemented in VEGA.

### 4.4. Statistical Analysis

Accuracy, sensitivity and specificity were calculated according to Baratloo et al. [71] to evaluate the classification models. For regression models, we calculated the r^2^, the RMSE and the MAE:(3)$$r^{2}=1-\frac{\sum_{i=1}^{n}{\left(y_{i}-{\hat{y}}_{i}\right)}^{2}}{\sum_{i=1}^{n}{\left(y_{i}-y_{avg}\right)}^{2}}=1-\frac{RSS}{TSS}$$
(4)$$RMSE=\sqrt{\frac{\sum_{i=1}^{n}{\left(y_{i}-{\hat{y}}_{i}\right)}^{2}}{n}}$$
(5)$$MAE=\;\left|y_{i}-{\hat{y}}_{i}\right|$$
where *y_i_* is the observed dependent variable (the experimental response), *ŷi* is the calculated value, *yavg* is the mean of the dependent variable, RSS is the residual sum of squares and TSS is the total sum of squares for *n* elements of the modeled dataset.

Finally, r^2^m metrics (including r^2^m, average r^2^m and Delta r^2^m) were calculated for validation purposes according to previously published approaches [72,73].

## 5. Conclusions

The protection of human health is the most important goal of public health management. The need to characterize the effects of chemicals is now considered a priority research area for environmental protection agencies and national institutes of health in different countries. In this study, we propose four QSAR models to assess chemical carcinogenicity, based on inhalation and oral slope factors, which are key parameters for health risk assessment, especially in the investigation of contaminated sites. To our knowledge, few models are available to quantitatively assess carcinogenicity, and most of them only predict in vivo oral carcinogenicity in animal models. Our combined approach can classify a compound as potentially carcinogenic or not, and it can estimate its carcinogenic potency for humans in terms of oral and inhalation slope factors in cases of carcinogenic activity.

The proposed models could help regulators to evaluate chemical substances for carcinogenicity in humans. Making a version of the models freely available will permit easy screening of chemicals, which will greatly support health risk assessments.

## Figures and Tables

**Figure 1 molecules-26-00127-f001:**
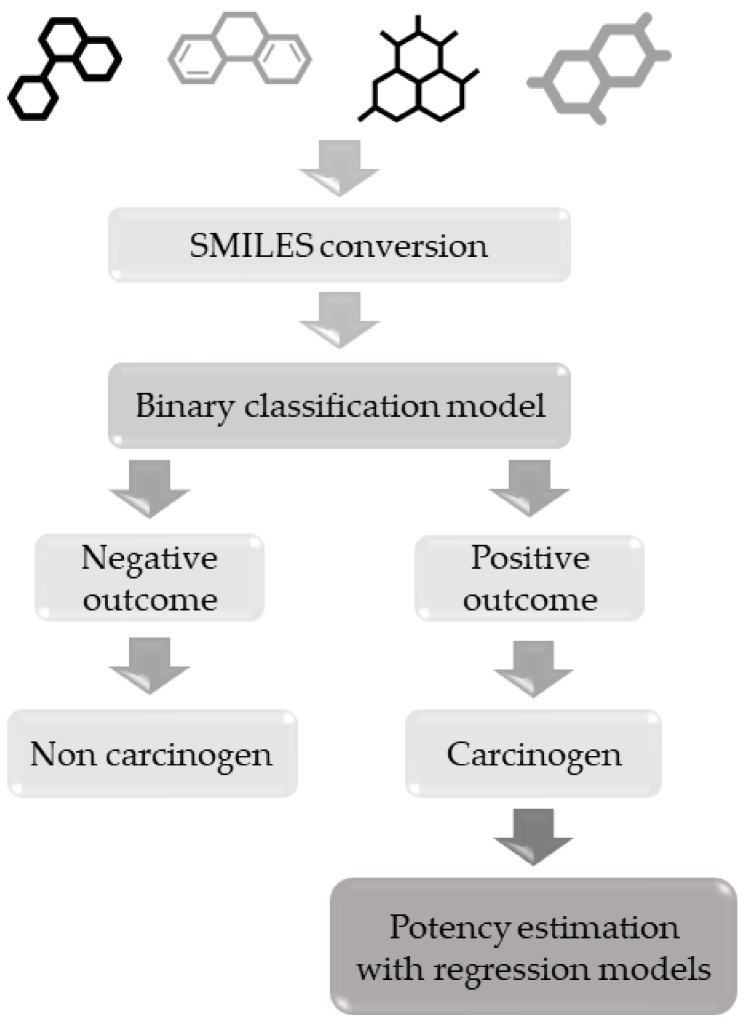
Hierarchical workflow to apply classification and regression-based models for carcinogenicity.

**Figure 2 molecules-26-00127-f002:**
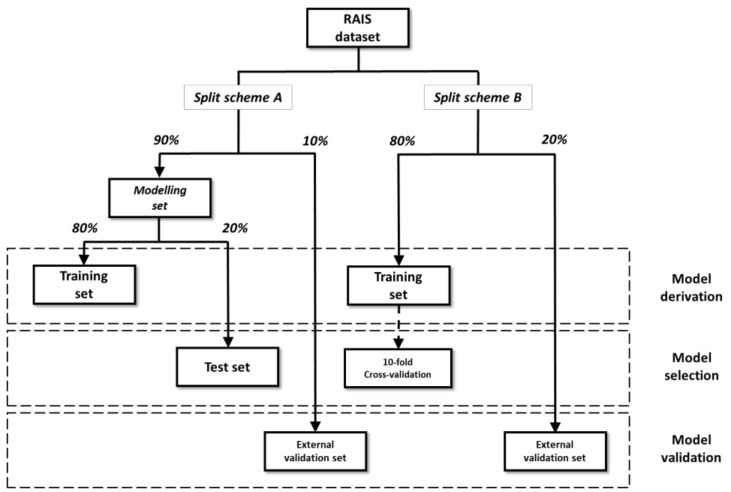
Graphic representation of the two schemes applied for dataset split. For each dataset, the function in model development (i.e., derivation, selection and validation) is reported.

**Table 1 molecules-26-00127-t001:** Statistics for the final oral and inhalation classification models (as implemented in VEGA).

Endpoint	Statistics	Training Set	Test Set
OSF	Accuracy	0.81	0.76
	Sensitivity	0.82	0.76
	Specificity	0.79	0.76
ISF	Accuracy	0.81	0.76
	Sensitivity	0.73	0.72
	Specificity	0.86	0.79

**Table 2 molecules-26-00127-t002:** Performance of OSF and ISF regression models derived with split scheme A.

Endpoint	Statistics	TrS	TeS	ES
OSF	r^2^	0.709	0.708	0.569
	MAE	0.651	0.784	0.946
	RMSE	0.875	0.945	1.255
	coverage	100%	81%	88%
ISF	r^2^	0.745	0.654	0.510
	MAE	0.639	0.875	0.933
	RMSE	0.876	1.035	1.176
	coverage	100%	85%	89%

**Table 3 molecules-26-00127-t003:** Performance of OSF and ISF regression models derived with split scheme B. Coverage is the percentage of compounds retained after applying the applicability domain (AD).

Endpoint	Statistics	TrS		ES
		Goodness-of-Fit	10-Fold Cross-Validation	
OSF	r^2^	0.756	0.608	0.515
	MAE	0.646	0.819	1.012
	RMSE	0.814	1.031	1.284
	coverage	100%	100%	76%
ISF	r^2^	0.745	0.591	0.647
	MAE	0.639	0.838	0.853
	RMSE	0.876	1.088	1.023
	coverage	n.a.	n.a.	81%

**Table 4 molecules-26-00127-t004:** Performance of OSF and ISF regression models after VEGA implementation. Performances are reported for the training and test sets. Since the applicability domain is evaluated with the ADI index (53), coverage is not reported.

Endpoint	Statistics	Train ^1^ Full	Train without Outliers	Test Full	Test in AD
OSF	r^2^	0.658	0.642	0.573	0.636
	MAE	0.713	0.725	0.951	0.881
	RMSE	0..96	0.969	1.28	1.131
	r^2^m	0.653	0.64	0.489	0.58
	r^2^mavg	0.655	0.633	0.501	0.595
	r^2^mDelta	0.004	0.015	0.023	0.029
	k	1.017	1.02	0.985	1.028
	ki	0.647	0.63	0.57	0.619
	F	536.384	477.804	40.315	48.974
	CCC	0.791	0.778	0.722	0.77
ISF	r^2^	0.586	0.577	0.566	0.592
	MAE	0.862	0.868	0.923	0.91
	RMSE	1.119	1.13	1.121	1.115
	r^2^m	0.584	0.574	0.524	0.525
	r^2^mavg	0.568	0.562	0.484	0.482
	r^2^mDelta	0.034	0.024	0.08	0.085
	k	0.993	0.996	0.916	0.936
	ki	0.595	0.582	0.63	0.635
	F	330.968	302.773	32.615	34.808
	CCC	0.74	0.732	0.729	0.738

^1^ The training set for the OSF is TrS + TeS.

## Data Availability

The data presented in this study are openly available in Zenodo at doi:10.5281/zenodo.4385768 and in VEGAHUB website at https://www.vegahub.eu/portfolio-item/repository-of-models-dataset/.

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
