# Peer review of "QSAR Models for Human Carcinogenicity: An Assessment Based on Oral and Inhalation Slope Factors"

_molecules, 2020, doi:10.3390/molecules26010127_

Round 1
Reviewer 1 Report
Whereas the work seems to be carefully done, some few points need attention before publication. Please find bellow some suggestions to improve the quality of the manuscript.
1. The authors should report applicability domain of the developed models according to Chemom Intell Lab Sys, 145, 2015, 22-29, and Chemical Biology & Drug Design, 87, 2016, 455-466.
2. The authors should use rm2 metrics for validation. See J Comput Chem 34, 2013, 1071-1082 and Journal of Chemistry, v. 2016, p. 1-12, 2016 (http://dx.doi.org/10.1155/2016/9198582). All suggested references should be included in the paper as well.
Author Response
Dear Reviewer 1, thank you for your helpful contribution. We worked to provide the information you required and to implement them into the text.
The applicability domain of the developed models was defined also according to the suggested approach. Results of this work can be found in the revised table 4 and in the sentence added at row 292-293.
R2m metrics for validation were calculated according to the proposed approach and reported in the revised table 4. Sentence in rows 315-316 was added to introduce the use of the suggested approach and the relative references.
Finally, the suggested references were included in the paper.
Reviewer 2 Report
The review of the paper "QSAR models for the human carcinogenicity: an assessment based on oral and inhalation slope factors".
The paper provides a systematic QSAR scheme based on a combination of classification and regression models that can be applied to assess the carcinogenicity potential of a range of organic compounds introduced via oral or inhalation tract.
I do not have general problems with the scientific side of the presented research. However, I do have multiple remarks concerning the way it is presented. Most of my reservations are provided as comments to the manuscript (attached to the review).
One general comment - one can see that the authors are not accustomed to the MDPI style which puts the methods section at the end of the manuscript (or the paper was re-edited to fit such scheme). As a result, the reader is confronted with a lack of knowledge on the meaning of abbreviations used or models developed in the study. This is a serious handicap of the paper which should be amended.
Also, I am not satisfied with the description of a final regression model. Maybe it is my personal trait as a specialist in ANN but I really would like to know something more about the developed models besides the number of the hidden neurons. The research in the modeling area should be also reproducible therefore I urge authors to deposit their curated database in some repository (e.g. DATA, or Mendeley data, Zenodo etc.) so their work can be reused. This FAIR approach should also increase the citation of the paper.

Author Response
Dear Reviewer 2, thank you for your helpful contribution. We worked to provide the information you required and to implement them into the text.
I apologize for the inconvenience you have pointed out about the use of acronyms. In the first submission we formatted the text according to the scheme required by the journal but we did not consider the order of introduction of the acronyms. Following his suggestion, we proceeded to re-examine the text, indicating in full each acronym at the first introduction.
We added an explicatory paragraph to descrive the ANN models in details. New paragraph is now at rows 275-285.
Furthermore, accepting the auditor's suggestion, we have uploaded the datasets to Zenodo to comply with the FAIR principles. The datasets are included in the paper as supplementary material and are already available on VEGAhub.